# Mitochondrial Functioning ≠ General Intelligence

**DOI:** 10.3390/jintelligence8020020

**Published:** 2020-05-03

**Authors:** Alexander O. Savi, Han L. J. van der Maas, Gunter K. J. Maris, Maarten Marsman

**Affiliations:** 1Department of Educational and Family Studies, VU University Amsterdam, 1081 BT Amsterdam, The Netherlands; 2Department of Psychology, University of Amsterdam, 1018 WS Amsterdam, The Netherlands; H.L.J.vanderMaas@uva.nl (H.L.J.v.d.M.); M.Marsman@uva.nl (M.M.); 3ACTNext, by ACT, Iowa City, IA 52245, USA; Gunter.Maris@act.org

**Keywords:** networks, formal theory, cyclic mechanisms, individual differences

Geary puts forward an appealing argument for the consideration of mitochondrial functioning as a candidate for a formative *g*
[1] ([1]); it is also an ambitious argument. After Spearman’s discovery of a psychometric *g* in 1904 [2] ([2]) and his speculations about a psychological *g* in 1927 [3] ([3]), researchers have been attracted by the simplicity of a single unifying factor. To date, the identification of this hypothesized factor is still awaited. In fact, various contemporary theories suggest that local complex dynamic mechanisms drive the emergence of the global phenomenon *g*
[4] ([4]); [5] ([5]); [6] ([6]); [7] ([7]). In the following, we comment on Geary’s attempt at debunking *g* and suggest a different—but nonetheless compatible—perspective. We also—not entirely accidentally—suggest the network perspective [8] ([8]) as a more principled method that enables the study of the influence of potentially powerful factors on intelligence in their biological, psychological, and societal context.

## 1. Hierarchies ≠ Unidirectional

One may indeed be tempted to think that a nested structure suggests that “deficits or inefficiencies at lower levels will ripple through all higher levels but deficits at higher levels …need not have broad influences at lower levels” [1] ([1]) (p. 2). Evidently, reality is more complex. As Sternberg pointed out in this very issue [9] ([9]), in the field of (behavioral) epigenetics many top-down mechanisms have been identified. More specifically, various mechanisms—including DNA methylation [10] ([10])—have been found to be involved with memory formation and storage, which is evidently linked to intelligence. We believe that as long as such mechanisms are still poorly understood, one must either be wary of very strict assumptions or establish evidence that mitochondrial functioning is truly independent of higher-level systems.

## 2. Biological Mechanisms ≠ Formative Factors

In addition, by suggesting that “mitochondrial functioning is the most fundamental biological mechanism”, [1] ([1]) (p. 1) seems to ignore influences on mitochondrial functioning by not just higher-level, but also same and lower-level systems; that is, the complex biological and psychological reality in which our mitochondria function. Such influences, whether unidirectional or cyclic, impede the determination of a formative factor or the “most fundamental mechanism”. Cyclic mechanisms can preeminently create chicken-or-the-egg dilemmas. In fact, cyclic mechanisms are abundant in biological systems such as in humans. We highly recommend Bechtel and Abrahamsen’s elaborate account of complex biological mechanisms [11] ([11]). In conclusion, we believe that one cannot simply assume that other (cyclic) mechanisms are either absent or negligible, and that mitochondrial functioning is indeed an independent causal mechanism.

## 3. Ceilings ≠ Ceiling Effects

Then, Geary suggests that “the capacity for mitochondria to produce energy places a ceiling on the performance of higher-level systems” [1] ([1]) (p. 3). This, again, is a strict assumption. A ceiling may never be reached, and if reached, lesser amounts of energy may also be sufficient for superb cognitive ability, with additional energy production providing no further contribution. Put differently, a ceiling does not guarantee a ceiling effect. Is there evidence that variation in the capacity for mitochondria to produce energy causes variation in intelligence? That is, does this ceiling ever reach a level that limits cognitive development? Additionally, are there no other factors that constrain or alter this capacity and that may be deemed equally important?

## 4. Verbal Theory ≠ Theory

Sternberg rightly remarks that correlation is not causation and that one should therefore be wary of correlation-backed causal claims [9] ([9]). In addition, we argue that verbal theory is not theory, but speculation. We believe that speculation propels scientific progress; however, only when approached with appropriate care and followed up with extensive verification. Ultimately, the field of human intelligence benefits most from formal theory. Formal theories explicate the exact hypothesized mechanisms from which the observed phenomena emerge and allow for precise predictions of still unobserved phenomena. Evidently, formal theories are by no means the end of scientific progress, if only because substantially different theories can turn out to be mathematically equivalent [12] ([12]). Indeed, model predictions must be verified, experiments can be used to validate causal claims, and time-series data may provide further proof.

## 5. General Intelligence ≈ Network

Geary’s “nested structure of brain systems” illustrates, on a high level, some of the factors that may be involved in the dynamic human systems from which the *g* phenomenon ultimately emerges. However, a common representation that does more justice to all possible complexities involved is a network [13] ([13],[8]). In a network, nodes are connected by edges to represent the active elements and their relations. Interestingly, the network perspective provides a principled method for exploring the problems discussed above.

In 2006, van der Maas and colleagues introduced the very first network approach to general intelligence [5] ([5]). Fundamentally, their model proved that psychometric *g* can simply emerge from cyclic psychological processes. In 2019, we showed how psychometric *g* emerges from the wiring of facts and procedures during cognitive development [7] ([7]). Crucially, our model respects individual differences in the diverse cognitive abilities that shape our intellect, and although rudimentary at present, the model serves as a basis for including biological, psychological, and even societal influences. Importantly, both models were developed within a thorough formal framework and obviate the need to find a single unifying factor that explains everything.

In conclusion, networks defy a race to the reductionistic bottom [14] ([14]), as they can help acknowledge the inevitable influence of functions at different levels of the human system. Indeed, multilevel networks allow for connecting theories at these different levels. To illustrate this, the theoretical foundation for the community structure in our—mostly psychological—model ([7]) is nicely laid out in Barbey’s network neuroscience theory of human intelligence [15] ([15]). Certainly, we also welcome Geary’s plea to consider mitochondrial functioning as a mechanism related to general intelligence. However, we invite researchers in the field of human intelligence to respect the system’s complex and dynamic nature, to approach it as an open rather than closed system, and to explore the proposed mechanisms in their rich biological, psychological, and societal context.

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
