# Peer review of "Mitochondrial Functioning ≠ General Intelligence"

_jintelligence, 2020, doi:10.3390/jintelligence8020020_

Round 1
Reviewer 1 Report
This is an excellent commentary, evaluating Geary's paper (and theory) as such and from the perspective of the authors' own theory. I found both approaches in the commentary very sound and eloquently presented in spite of the short space available, even if one might not accept the authors' theory in its entirety. I believe that the commentary will give Geary the opportunity to defend his theory in a rejoinder, focusing on the arguments advanced here. Therefore, I propose that the commentary is accepted for publication as is.
Author Response
We thank the reviewer for her or his positive review.
Reviewer 2 Report
The authors present five arguments that highlight the limitations of the reductionist approach proposed by Geary (2019).
The fifth argument seems to me the most important because it opens more perspectives. The authors may be encouraged to go a little further. More precisely it could be interesting to take the opportunity of abording the notion of multilevel network in prolongation of their multilevel framework “biological, psychological, and societal” (l. 72).
Linking intelligence directly to the functioning of the mitochondria is like linking the performance of cars directly to their fuel consumption. This does not explain how the system works, which is however the main goal of the study of intelligence.
The network perspective is by far more promising when it comes to describe a mechanism rather than predict its outcomes. It makes it possible to understand the link between two levels of observation, each level having its own network of process. The mitochondria are the site of a mechanism, the Krebs cycle, which involves a set of interacting elements and participate in a more general mechanism at the intra-cell level. The cells themselves, which can be neurons, interact with each other in a larger network and so on. A multi-level approach whose aim is to contribute to the understanding of the mechanisms can hardly do without a description of the networks at each level of observation.
Author Response
We thank the reviewer for her or his positive review. To illustrate how network theories that describe different levels can be integrated, we added the following sentence to the fifth argument:
"Indeed, multilevel networks allow for connecting theories at these different levels. To illustrate this, the theoretical foundation for the community structure in our—mostly psychological—model [7, pp. 12–13], is nicely laid out in Barbey’s network neuroscience theory of human intelligence [15]."
Indeed, multilevel68networks allow for connecting theories at these different levels. To illustrate this, the theoretical69foundation for the community structure in our—mostly psychological—model [7, pp. 12–13], is70nicely laid out in Barbey’s network neuroscience theory of human intelligence [15].
Round 2
Reviewer 2 Report
I have no more comment.